# Inhaled Nitric Oxide Promotes Angiogenesis in the Rodent Developing Brain

**DOI:** 10.3390/ijms24065871

**Published:** 2023-03-20

**Authors:** Gauthier Loron, Julien Pansiot, Paul Olivier, Christiane Charriaut-Marlangue, Olivier Baud

**Affiliations:** 1Service de Médecine Néonatale et de Réanimation Pédiatrique, Université de Reims Champagne-Ardenne, CReSTIC, CHU Reims, 51100 Reims, France; 2Inserm, NeuroDiderot, Faculty of Medicine, Université Paris Cité, 75019 Paris, France; 3Division of Neonatology and Pediatric Intensive Care, Children’s University Hospital of Geneva, 1205 Geneva, Switzerland

**Keywords:** nitric oxide, developing brain, angiogenesis

## Abstract

Inhaled nitric oxide (iNO) is a therapy used in neonates with pulmonary hypertension. Some evidence of its neuroprotective properties has been reported in both mature and immature brains subjected to injury. NO is a key mediator of the VEGF pathway, and angiogenesis may be involved in the reduced vulnerability to injury of white matter and the cortex conferred by iNO. Here, we report the effect of iNO on angiogenesis in the developing brain and its potential effectors. We found that iNO promotes angiogenesis in the developing white matter and cortex during a critical window in P14 rat pups. This shift in the developmental program of brain angiogenesis was not related to a regulation of NO synthases by exogenous NO exposure, nor the VEGF pathway or other angiogenic factors. The effects of iNO on brain angiogenesis were found to be mimicked by circulating nitrate/nitrite, suggesting that these carriers may play a role in transporting NO to the brain. Finally, our data show that the soluble guanylate cyclase/cGMP signaling pathway is likely to be involved in the pro-angiogenetic effect of iNO through thrombospondin-1, a glycoprotein of the extracellular matrix, inhibiting soluble guanylate cyclase through CD42 and CD36. In conclusion, this study provides new insights into the biological basis of the effect of iNO in the developing brain.

## 1. Introduction

Nitric oxide is a ubiquitous and key player in many biological processes, from embryogenesis to aging [1]. NO regulates regional blood flow by relaxing the arteriolar smooth muscle via guanylate cyclase and cGMP production [2]. Because of its vasodilatory properties, the use of inhaled NO (iNO) has been approved by the Food and Drug Administration for the treatment of persistent pulmonary hypertension in newborns [3,4]. In addition, NO is involved in physiological and pathological angiogenesis (tumorigenesis) via the signaling pathway linked to the vascular endothelial growth factor receptor (VEGFR). A defective VEGF pathway and angiogenesis blockade were found to be associated with bronchopulmonary dysplasia (BPD), a common condition associated with very preterm birth [5]. Therefore, iNO was tested as a potential candidate to prevent chronic lung disease in very preterm infants. While animal studies were promising and confirmed the ability of iNO to promote angiogenesis in developing lungs, prophylactic or therapeutic exposure to NO failed to prevent BPD in human neonates [6].

As a highly reactive radical, iNO was expected to react very rapidly and locally in the pulmonary capillary circulation with other reactive species, theoretically precluding any remote effect in other organs. However, several lines of evidence have demonstrated that iNO is associated with neuroprotective effects both in mature and developing rodent brains [7,8,9]. Indeed, iNO was shown to improve blood flow and neuronal survival following ischemic conditions in the adult brain. In the developing brain, we previously reported a beneficial effect of iNO on brain damage induced by either excitotoxic, oxidative, or hypoxic-ischemic injury [8].

These biological effects observed in the brain are expected to be mediated by NO transport in a bioactive form, which has already been shown in previous studies [7]. Among carrier molecules, S-nitrosothiols, N-nitroso-hemoglobin, and nitrite enable NO release via an oxygen-tension-dependent mechanism [10]. However, how iNO can confer neuroprotection remains unclear. In addition to the effect of NO on cerebral blood flow, we also reported that iNO enhances oligodendroglial lineage maturation [11]. Postnatal white matter angiogenesis has been shown to be critical for the onset of myelination and axon integrity through hypoxia-inducible factor (HIF) signaling [12]. Therefore, we hypothesized that iNO may promote angiogenesis in the developing brain, leading to the reduced vulnerability of white matter and cortex to injury. Here, we investigated the effect of iNO on angiogenesis in the developing brain and its potential effectors.

## 2. Results

### 2.1. Inhaled NO Promotes Angiogenesis in the Developing Brain during a Critical Window

Exposure of rat pups to inhaled nitric oxide (iNO) from E21 to P7 was associated with a significant increase in vascular density in P14 rat pups, as revealed by the quantification of MCT-1 immunoreactive vessel walls (Figure 1). In the control animals, vascular density was found to be similar at P7 and P14, and a two-fold increase was observed between P14 and P21. In animals exposed to iNO, vascular density increased significantly earlier, starting at P7 and reaching a vascular density similar to that observed in control animals at P14 one week later (Figure 1A–C). The developmental shift in vascular density was similarly observed in both the cingulate cortex and white matter and was found to be consistent using quantification of the vessel density and intersection of vessels with lines of a 10 × 10 grid at ×400 magnification (as a marker of vessel branching, see Methods). No difference was detected between 5 ppm and 20 ppm of iNO exposure. Therefore, only 5 ppm of iNO was used in subsequent experiments. Because exposure to prolonged or high levels of iNO may induce methemoglobinemia and impair oxygen delivery to tissues, we analyzed the arterial blood gas in rat pups. No statistical difference was detected in methemoglobin and PaO_2_ between animals exposed to room air or iNO (Appendix A). Despite a slight decrease in PaCO_2_ observed only at the onset of NO exposure, the blood flow velocity in the basal truncus assessed by Doppler was previously found to be unchanged compared to animals exposed to room air [9].

### 2.2. The Effects of iNO on Angiogenesis Are Mimicked by Nitrate/Nitrite

Perinatal and early postnatal exposure to iNO was associated with a significant 2.6-fold increase in nitrate/nitrite (nOx) blood concentrations in P1 pups (Figure 2A). To assess the potential role of nOx in the pro-angiogenic phenotype of iNO, we first monitored serum nOx concentrations following an i.p. injection of 40 mg/kg of sodium nitrite. We found that the kinetics of nOx concentration within the 12 h following nitrite injection were highly comparable to the nOx serum concentration pattern measured with 5 ppm of iNO exposure (Figure 2B). We next performed MCT-1 immunolabeling in animals exposed to i.p. nitrite from E21 to P7 and found a similar significant increase in vascular density in the cingulate cortex at P14 (Figure 2C).

### 2.3. Inhaled NO Effects in the Developing Brain Do Not Engage Changes in Endogenous NO Synthesis

We next assessed the potential role of endogenous NO production in the pro-angiogenic phenotype conferred by exogenous iNO (Figure 3). Endothelial and neuronal nitric oxide synthase (eNOS/NOS3 and nNOS/NOS1, respectively) are responsible for constitutive NO production within the developing brain. Both gene expression, assessed by qPCR, and protein concentrations, assessed by Western blot, remained unchanged by iNO exposure at both P7 and P14 (Figure 3A–D).

To further demonstrate that the effect of iNO on angiogenesis is not related to endogenous NO production, we blocked all NO synthases from P0 to P7, using N omega-nitro-L-arginine methyl ester hydrochloride (L-NAME), a cell-permeable NOS inhibitor. As a result of L-NAME treatment, we observed a severe postnatal growth restriction lasting at least 7 days. In contrast, growth failure was found to be prevented in animals exposed to L-NAME but rescued by exogenous 5 ppm iNO (Appendix A). Vascular density enhancement induced by iNO at P14 was not prevented by the concomitant L-NAME, as revealed by the quantification of MCT1-immunoreactive vessel walls in the cingulate cortex (Figure 3E). These data confirm that NOS modulation by exogenous iNO is unlikely to account for its effect on brain angiogenesis. 

### 2.4. Inhaled NO Effects in the Developing Brain Are Not Mediated by NO-Related Angiogenic Factors

Because the VEGF/VEGFR signaling pathway is a key player in angiogenesis, we next examined whether the effect of iNO on vascular density in the developing brain could be associated with changes in the expression of genes encoding for VEGFa and VEGFR2 or VEGF concentration in brain tissue. VEGFa and VEGFR2 gene expression was not found to be modified by iNO at any of the ages studied (Figure 4A,B). VEGF protein concentration in the forebrain, assessed by ELISA, was only slightly decreased in P1, but not in P7 and P14 animals exposed to iNO, compared to the controls (Figure 4C).

To further confirm the lack of involvement of the VEGFa/VEGFR2 pathway in the proangiogenic effect of iNO, we selectively blocked VEGFR2 using SU5416. SU5416 exposure induced no change in the iNO-related increase in vascular density, as assessed by MCT1 immunoreactivity in P14 animals in both the cortex and the developing white matter at P14 (Figure 4D).

The gene expressions of other factors recognized to control angiogenesis, assessed by qPCR, were found to be similar between air- and iNO-exposed animals (Appendix A).

### 2.5. Evidence for the Role of Soluble Guanylate Cyclase/cGMP Pathway

As an increased cGMP concentration in brain tissue has been reported in response to iNO [13] (Figure 5A), we hypothesized that a modulation of soluble guanylate cyclase (sGC) may be involved in the proangiogenic effect of iNO. Thrombospondin-1, a glycoprotein of the extracellular matrix, was described as an inhibitor of NOS and sGC through CD42 and CD36, respectively (Appendix A). We found TSP1 expression to be downregulated at both P7 and P14, with a significant effect observed in iNO-exposed animals only at P14 (Figure 5B). Finally, we further confirmed that the sGC/cGMP signaling pathway is involved by using ABT510, a specific CD36 agonist. While i.p. injection of ABT510 from E21 to P7 did not change vascular density in room air-exposed animals, it significantly abolished the pro-angiogenic effect of iNO in both the cingulate cortex and white matter at P14 (Figure 5C). 

## 3. Discussion

This study shows that iNO induces a shift in angiogenesis in the developing brain of rat pups, with an increased vascular density earlier in the brain development. NO synthases, VEGF, and other canonical factors involved in the regulation of angiogenesis do not appear to mediate this effect. In contrast, the regulation of cGC by TSP-1 plays a role in our rodent model. 

Using the same paradigm, we previously published that iNO promotes myelination at P7 in the developing brain through an advance of oligodendrocyte lineage maturation and myelin production [11]. This effect is notably mediated by increased PDGFR-alpha expression [11]. Similarly, iNO is associated with the proliferation of neural progenitor cells subsequently differentiating as oligodendrocytes and pericytes [13]. Taken together, these results suggest a broad pro-maturative effect of exogenous, inhaled NO in the developing rodent brain.

### 3.1. Signaling Pathways Involved in iNO Angiogenic Effects in the Developing Brain

Assessing NO levels in brain of living animals is challenging. Most detection methods are based on oxidation-reduction reactions that do not differentiate NO from its derivatives: free radicals, nitrite/nitrate, or even S-nitrosylated proteins [14]. Using in vivo voltametric detection, we previously reported that exposure to 20 ppm of iNO was associated with a 140% significant increase in cortical concentration of the NO molecule in P7 rats [9]. However, the release of NO in animals exposed to iNO is often inferred from the observation of its hemodynamic or biological effects, as we and others have previously reported [7,9]. 

Given that iNO was associated with an increase in NO in the brain, the next step was to identify which factors mediate its effect on angiogenesis. VEGFa and its main receptor, VEGFR2, are key players of angiogenesis during development [15,16]. However, the body of evidence of the present study supports that iNO’s impact on brain development is not mediated by VEGF or NOS activation. Our data are consistent with a replacement effect of iNO when NOS are inhibited by L-NAME, as reported in a stroke model in rat pups [9]. 

We found that the use of a thrombospondin-1 (TSP-1) agonist was sufficient to abrogate the iNO-induced phenotype. TSP-1 has been described as a “gatekeeper” of nitric oxide signaling [17]. It is a large glycoprotein found in the extracellular matrix and is involved in inflammation, blood flow regulation, and tumorigenesis [18,19]. More specifically, among many TSP-1 receptors, CD47 and CD36 exert an anti-angiogenic effect via VEGFR2-linked proteins, endothelial NOS, and sGC [20,21]. Here, we found that TSP-1 was downregulated at P14 in animals exposed to iNO. Because the TSP-1 agonist, ABT510, which targets CD36, prevented the NO-induced pro-angiogenic phenotype, sGC activation may be the prominent pathway involved in the iNO-related pro-maturative effects in the developing brain.

### 3.2. Remote Effect of iNO on Peripheral Organs

The critical question regarding our experiments is whether exogenous NO can reach the brain or other peripheral organs. In addition to the effect of iNO on cerebral blood flow in the neonatal brain, Kuebler et al. documented cerebral vasodilation in rat and pig brains during inhalation of NO at 5, 10, or 50 ppm [22]. However, the effects of iNO on vasoreactivity and blood flow are not restricted to the brain. Hence, iNO has been shown to remotely increase or restore blood flow in the kidney [23], skin, and bowel [24,25]. As a reactive species, NO rapidly interacts with many ligands, and iNO is very likely to be carried in the lungs before any exposure of the peripheral organ can occur [26]. Two metabolic fates are well-recognized for endogenous NO. First, guanylate cyclase, the cytoplasmic canonical target of NO, generates cyclic GMP then activates cGMP-protein kinase and cGMP-dependent signaling [27,28]. Second, nitric oxide redox reacts with thiol radical L-cystein in a reaction of S-nitrosylation. More recently, enzymatic complexes have been described that trigger nitrosylation and de-nitrosylation [29,30], suggesting a regulated biological activity [31]. S-nitrosylation may account for large post-transcriptional regulation [32,33]. 

Due to the very high affinity of sGC for its natural ligand, remotely delivered iNO may interact with sGC rather than engaging S-nitrosylation [34,35]. Indeed, we have reported a more than four-fold increase in cGMP concentration in brain tissue extracted just one hour following iNO exposure in P1 rat pups, suggesting a rapid activation of sGC [13]. Consistently, serum cGMP concentration was also found to be increased after exposure to 80 ppm of iNO in lambs [36]. 

Because the hemoglobin level was slightly higher at P7 after NO exposure compared to the control, we cannot rule out that increased oxygen delivery may play some role in the increased vessel density in the cortex and white matter.

### 3.3. iNO and Angiogenesis: A Role in the Neuroprotective Effect of the NO Pathway in Preclinical Models of Brain Injury?

It has been speculated that the NO pathway may provide some neuroprotection in preclinical models of brain injury. Neuroprotective effects of prophylactic iNO exposure in different models of excitotoxicity in rat pups have been reported, mediated by a modulation of the expression of NMDA receptor subunits (pCREB) [8]. Such a neuroprotective effect was also demonstrated in both hyperoxic and hypoxic models of brain damage [37,38,39]. iNO reduces the inflammatory response and TNF signaling at low doses [40,41]. The main biological effect of NO—an increase in cGMP concentration—can also be achieved by sildenafil, a highly potent selective inhibitor of phosphodiesterase type-5 (PDE5i). Similarly to iNO, it was reported to induce neuroprotective and neurorestorative properties, particularly in the neonatal hippocampus in neonatal models of hypoxic ischemic brain insult [42,43,44]. In addition to its effects on vascular vulnerability and cerebral hemodynamics, sildenafil has been shown to provide anti-inflammatory effects and protection against lesion extension in the late phase after pMCAo in neonatal mice [45,46]. Highly consistent data have been reported in the mature brain in both rodent and porcine models of traumatic brain injury [47,48]. Again, the modulation of microglial activation, inflammatory cascade, and oedema appear to be the most relevant targets of NO exposure [48,49]. These effects on neuroinflammation were found to be associated with improved short-term memory [49]. 

### 3.4. NO Pathway: A New Neuroprotective Strategy in Human Neonates?

Preterm birth and perinatal brain injury expose infants to neurological complications and neurodevelopmental impairments. These abnormalities are partly the result of an amalgam of challenging conditions experienced by preterm infants, including inflammation and hypoxic-ischemic insult, during a critical developmental window of vulnerability [50]. Notably, systemic inflammation and subsequent microglial activation result in disruption of the oligodendrocyte lineage, abnormal myelination, and the global impairment of brain maturation [51]. 

However, in human preterm neonates, no beneficial effect of iNO exposure (tested in controlled clinical trials for many reasons other than neuroprotection) has been observed so far on long-term neurological outcomes, regardless of the dose and timing of exposure [52]. The multifactorial nature of the damage to the developing brain with temporal and spatial heterogeneity, the duality of NO according to the dose and the cellular context, and the absence of cell targeting and confounding morbidities indicating iNO exposure are all factors that may participate in this discrepancy between animal models and clinical research. In addition, the involvement of the NO pathway in modulating the extent/repair of brain damage following a stroke is complex. In the early phase of reduced cerebral blood flow, NO allows the recruitment of collaterals in the penumbral zone. On the other hand, in the reperfusion phase, NO synthesis by inducible or neuronal NOS is deleterious, exacerbating apoptosis, inflammation, and free radical synthesis [46,53,54,55]. Furthermore, the role of each NOS is inhomogeneous depending on the isoform, brain region, and time point.

### 3.5. Conclusions

In conclusion, this study provides new insights into the biological basis of the effect of iNO in the developing brain. Our data support an effect of iNO on angiogenesis in the developing brain through blood carriers, leading to the modulation of the cGMP pathway and, more specifically, regulated by TSP-1. This effect may explain, at least partly, the effect of iNO on myelination and its neuroprotective properties previously documented in rodent models of perinatal brain injury. 

## 4. Materials and Methods

### 4.1. Animals, NO Exposure, and Brain Sampling

All animal experiments were performed in accordance with INSERM ethical guidelines, and the animal study protocol was approved by the Robert Debré Hospital Research Council Review Board (A75-19-01). Female rats (Sprague Dawley; Janvier, Le Genest St. Isle, France) and their pups were placed in a normoxic, normocapnic gas chamber containing 5 ppm or 20 ppm of NO from E21 (the day before the expected delivery) to P7. An iNOvent^®^ system (Ikaria, Clinton, NJ, USA) continuously monitored NO and NO_2_ concentrations. Apart from exposure to NO, the rear conditions were the same in both groups. To control for any maternal-mediated effects (e.g., milk), we rotated dams between cages twice a week without moving the rat pups. A priori sample size calculation indicated a minimum of 6–8 animals per group. Since exposure started on the last theoretical day of gestation, we assigned one to two pregnant rats and their litter (usually at least twelve pups) per experimental condition and assessed one to two timepoints per litter.

At each timepoint (P1, P7, P14), 6 to 10 animals were anesthetized with inhaled isoflurane (Abbott France, Rungis, France). Whole brains were either (i) immersed in 4% formol for 5 days, embedded in paraffin, and cut into 10 µm thick sections for immunohistochemistry; or (ii) immediately frozen in liquid nitrogen for molecular biology after transcardiac perfusion of the anesthetized rat pups with 4% paraformaldehyde in phosphate buffer (0.12 mol/L). Blood samples were collected by decapitation, and plasma was extracted after 20 min of decantation on ice and 15 min of centrifugation at 1500 rpm. A total of two hundred and fifty rat pups were used in this study. We did not observe any mortality induced by exposure to nitric oxide. 

### 4.2. Drugs

The VEGF pathway was assessed using the VEGF-R inhibitor SU 5416 (Tocris, Bristol, UK); 10 mg/kg was intraperitoneally injected (i.p.) twice daily from E21 to P7. To investigate the role of NO synthases, we inhibited all isoforms with N(G)-nitro-L-arginine methyl ester (L-NAME; 15 mg/kg was administered twice daily via i.p. injection from E21 to P7). To investigate the role of thrombospondin-1 (TSP-1), we used a pharmacological agonist of CD36, ABT510, which was developed by Abbott (Abbott Park, US-IL) and generously provided under transfer agreement N°45068. A dose of 100 mg/kg was injected i.p. every 12 h from E21 to P7.

### 4.3. Immunohistochemistry and Vessel Wall Density Quantification

All quantitative measurements were performed by two observers blinded to the experimental groups. Vessels were labeled with primary antibodies targeting monocarboxylate transporter 1 (MCT1 1:1000, Chemicon, Temecula, CA, USA), revealed using the streptavidin-biotin-peroxydase method (Jackson Immunoresearch laboratories, West Grove, PA), as previously reported [56]. MCT1 results were confirmed using tomatolectin immunolabeling (TL 1:500, Vector Lab, Burlingame, CA, USA); however, due to the better specificity of MCT1 for mature vessels, subsequent vessel quantification was based on the latter. The density of the vessels was assessed in the cingulate cortex and underlying white matter (+2.16 to 20.36 mm from the bregma) on four continuous sections, alternatively in the right or left hemisphere at P7, P14, and P21 using a 10 × 10 grid at ×400 magnification and covering an area of 0.065 mm^2^. Quantification was carried out in two ways: (i) quantification of immunoreactive vessels within the area of the grid; (ii) number of the vessels’ intersections with the grid’s lines, excluding intersections with the lower and left boundaries to appreciate the length and branching of the vessels [57].

### 4.4. Physiological Parameters

Arterial blood was collected by intracardiac puncture and physiological parameters were measured under air (*n* = 6) and 5 ppm of iNO exposure (*n* = 6).

### 4.5. ELISA Assays

VEGF quantification was performed using a specific ELISA in PBS-perfused brain samples according to the manufacturer’s instructions (R&D system Europe Lille, France). An enzyme immunoassay was used to quantify guanosine 3′,5′ cyclic monophosphate (cGMP) in the forebrain according to the manufacturer’s instructions (Cayman Chemical Company^®^, Ann Arbor, MI, USA). To assess the role of nitrite/nitrate (NO_2_/NO_3_) in the pro-angiogenic effect of iNO, rat pups received an i.p. injection of 40 mg/kg sodium nitrite every 8 h, from E21 to P7. A kinetic study was performed after a single injection of sodium nitrite. The forebrain cortex and white matter were homogenized and then ultra-centrifugated at 100.000 g in ultrapure PBS for 30 min, and nitrate/nitrite were assessed in supernatant using a chemiluminescent kit (Cayman Chemical Company^®^, Ann Arbor, MI, USA). All the above quantifications were adjusted for the protein concentration in the sample.

### 4.6. Western Blot 

Membrane proteins were extracted from the forebrain cortex, including the white matter, of P14 rat pups. Extraction was performed by homogenization in HEPES buffer containing protease inhibitors (Sigma, St. Louis, MO, USA) according to the manufacturer’s instructions. The protein samples (30 μg) were incubated overnight with either a NOS1, NOS3, or actin antibodies (see antibody Appendix A). Western blot experiments were run in triplicate with four animals per group.

### 4.7. Quantitative Real-Time Polymerase Chain Reaction

DNA-free total RNA from the brain cortex, including white matter, was obtained at P1, P7, and P14 using a previously reported protocol [11]. To standardize gene expression across the samples, we first compared the expression levels of four housekeeping genes within the samples. For reverse transcription (RT), we used 600 ng of total RNA and the iScript cDNA Synthesis Kit (Bio-Rad, Hercules, CA, USA). Real-time polymerase chain reactions (PCRs) were performed with Supermix (Bio-Rad) containing SYBR green for 50 cycles using a 3-step program [58]. Each reaction was run twice with a minimum of six animals per group; in both cases, samples were assessed in triplicate. The primers used for real-time PCR are listed in Appendix A.

### 4.8. Statistical Analysis 

In all experiments, the data are presented as mean ± standard error of the mean (SEM), and *p* < 0.05 was considered significant. A two-tailed unpaired Student’s *t*-test was performed for comparisons between two groups. For comparisons with more than two groups, either a one- or two-way analysis of variance was performed, and the Tukey post-hoc test was used. There were no a priori exclusion criteria. Outliers were identified using Grubb’s method. When appropriate, the normality of the distribution was checked using a D’Agostino–Pearson test or a Shapiro–Wilk test, depending on the number of values. Statistical tests were performed using GraphPad Prism version 9.5.1 (GraphPad Software, San Diego, CA, USA).

## Figures and Tables

**Figure 1 ijms-24-05871-f001:**
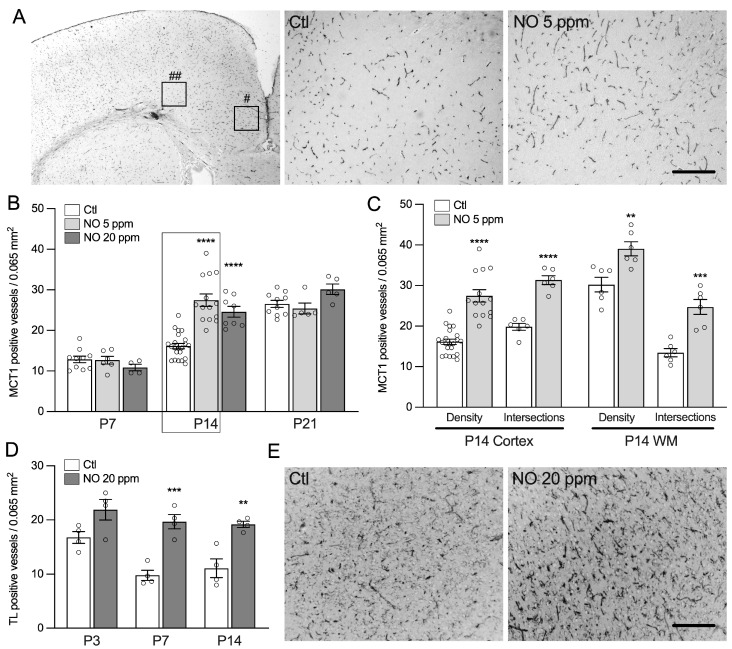
Inhaled NO enhances angiogenesis in the rodent developing brain. (**A**) Representative microphotographs of monocarboxylate transporter 1 (MCT1)-positive vessels, quantified in the cingulate cortex (#) and white matter (##). The two insets on the right show details in cingulate white matter from room air control (Ct) and rat pups exposed to 5 ppm of NO at P14 (×200). Scale bar = 250 µm. (**B**) Quantification of MCT1-positive vessels in cingulate cortex at P7, P14, and P21. One-way ANOVA for each age: **** compared to Ctl means *p* < 0.0001. (**C**) Quantification of MCT1-positive vessels in the cingulate cortex and cingulate white matter (WM) at P14, using quantification of vessel density and vessel intersections with grid lines. Student’s *t*-test: **, ***, and **** compared to Ctl mean *p* < 0.01, <0.001, and <0.0001, respectively. (**D**) Quantification of Tomatolectin (TL)-positive vessels in cingulate cortex at P3, P7, and P14. Student’s t-test: ** and *** compared to Ctl mean *p* < 0.01 and <0.001, respectively), in animals exposed to either room air control (Ctl) or 20 ppm of NO. (**E**) Representative microphotographs of TL-positive vessels in cingulum of control and NO-exposed P14 animals (×200). Scale bar = 250 µm.

**Figure 2 ijms-24-05871-f002:**
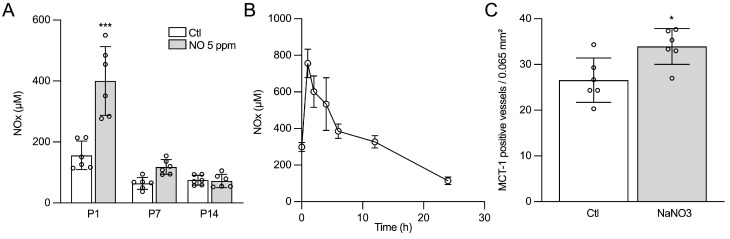
Nitrite infusion mimics inhaled NO’s pro-angiogenic effect. (**A**) Measurement of nitrite/nitrate (NOx) serum concentration in response to iNO exposure. Two-way ANOVA and Tukey post-hoc test: *** compared to Ctl means *p* < 0.001. (**B**) NOx kinetics after one 40 mg/kg i.p. injection of sodium nitrite. (**C**) Quantification of MCT1-positive vessels in cingulate cortex at P14 following 40 mg/kg/8 h i.p. injection of sodium nitrite from E21 to P7. Student’s *t*-test: * means *p* < 0.05.

**Figure 3 ijms-24-05871-f003:**
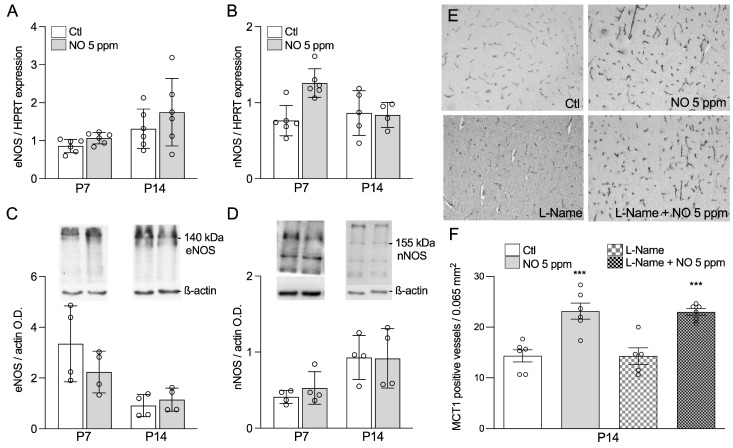
NOS gene expression in response to inhaled-NO exposure. eNOS (**A**) and nNOS (**B**) gene expression assessed using qPCR at P7 and P14 in animals exposed to room air (Ctl) or 5 ppm iNO from E21 to P7. eNOS (**C**) and nNOS (**D**) protein expression assessed using Western blot at P7 and P14 in animals exposed to room air (Ctl) or 5 ppm iNO from E21 to P7 (optical density relative to actin). (**E**) Representative micrographs of MCT1 immunolabeling in cingulate cortex at P14 of animals treated with NOS inhibitor L-NAME or saline and exposed to either room air or 5 ppm iNO (×200). (**F**) Quantification of MCT1-positive vessels in cingulate cortex at P14 of animals treated with NOS inhibitor L-NAME or saline and exposed to either room air or 5 ppm iNO. Two-way ANOVA: *** compared to Ctl means *p* < 0.001.

**Figure 4 ijms-24-05871-f004:**
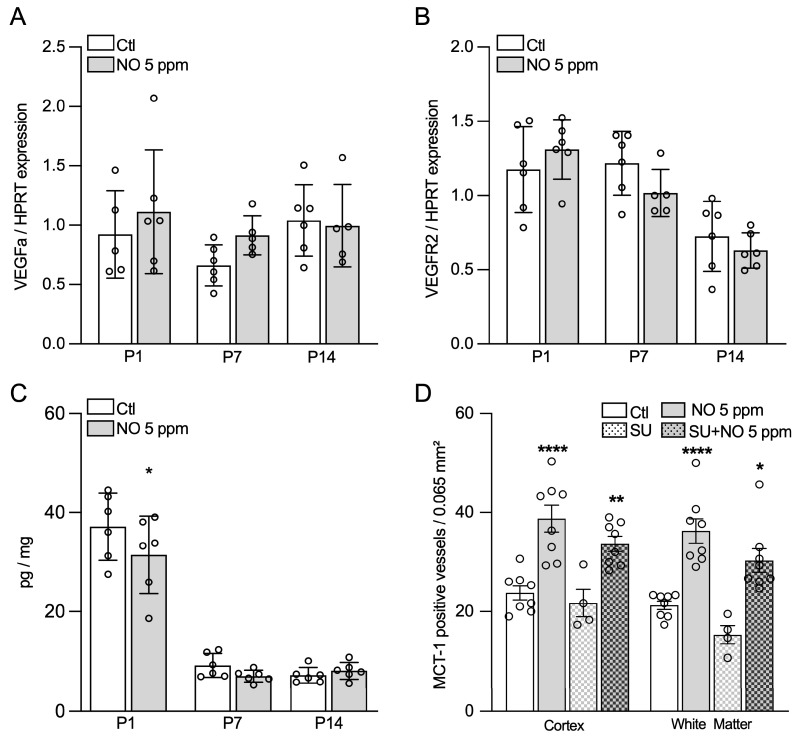
Inhaled nitric oxide pro-angiogenic effect does not involve VEGF. (**A**,**B**) VEGFa and VEGFR2 gene expression assessed using qPCR at P1, P7, and P14, respectively, in animals exposed to room air (Ctl) or iNO 5 ppm. (**C**) Elisa VEGF from forebrain extract, adjusted for protein concentration at P1, P7, and P14 in animals exposed to room air (Ctl) or iNO 5 ppm. (**D**) Effect of VEGFR2 blockade with SU5416 injection on MCT1-positive vessel quantification in cingulate cortex and white matter in P14 animals exposed to room air (Ctl) or iNO 5 ppm. Two-way ANOVA and Tukey post-hoc test: *, **, and **** mean *p* < 0.05, <0.01, and <0.0001, respectively.

**Figure 5 ijms-24-05871-f005:**
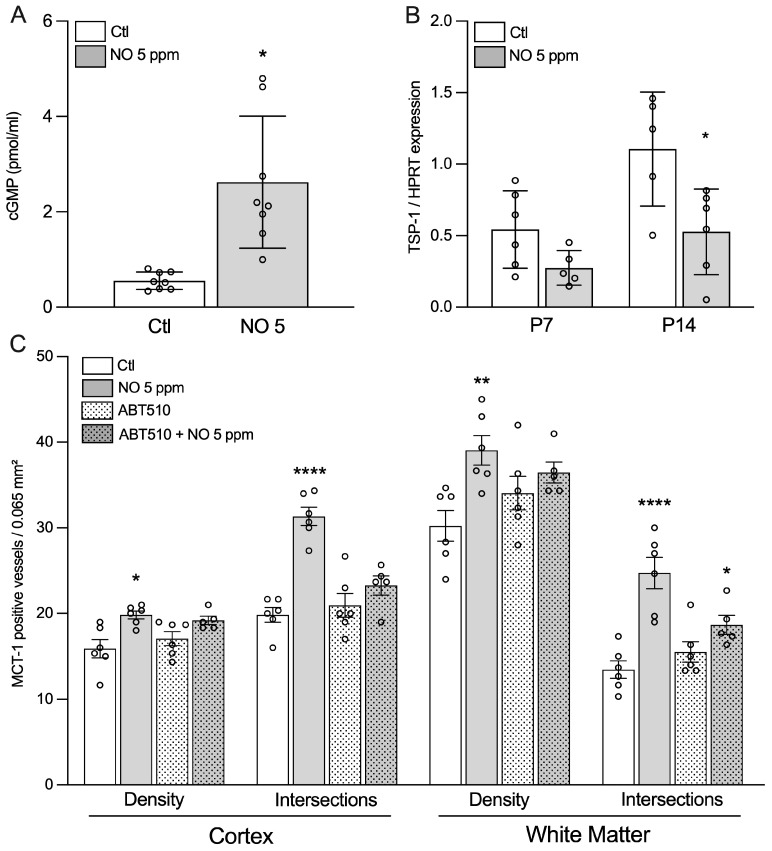
Evidence for release and intracellular action of NO molecule itself. (**A**) Inhalation of 5 ppm NO is associated with 5x increase of cGMP in white matter extract. Student’s *t*-test: * means *p* < 0.05. (**B**) Thrombospondin-1 (TSP-1) gene expression assessed using qPCR in P14 rat pups exposed to room air (Ctl) or iNO 5 ppm. Two-way ANOVA and Tukey’s post-hoc test: * compared to Ctl means *p* < 0.05. (**C**) Injection of a TSP-1 agonist, ABT510, significantly abolished the pro-angiogenic effect (as assessed using MCT1 immunolabeling) of iNO 5 ppm in both cingulate cortex and white matter. Two-way ANOVA and Tukey’s post-hoc test: *, **, and **** compared to Ctl mean *p* < 0.05, <0.01, and <0.0001, respectively.

## Data Availability

The data presented in this study are available on request from the corresponding author.

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
