# Peer review of "Inhaled Nitric Oxide Promotes Angiogenesis in the Rodent Developing Brain"

_ijms, 2023, doi:10.3390/ijms24065871_

Round 1

Reviewer 1 Report

In this manuscript, the authors demonstrate perinatal exogenous inhaled nitric oxide (iNO) promote angiogenesis in the developing white matter and cortex, and linked their findings to altered sCG/cGMP signalling by involvement of thrombospondin-1, validated by various pharmacological interventions.

This is an interesting report that presents useful information for readers to understand biological effects of iNO. Most parts are solid and well written, and quality is good. However, the present manuscript has some points that impair the originality of this paper. This paper needs few revisions to clear up the points listed below, which will improve the work.

i.           Did the authors observe differences in duration of pregnancies or litter size, was there a difference in breathing frequencies and cerebral blood flow in iNO-treated pups, as arterial CO2 was decreased in treated animals at earlier time points? Please check wording in lines 74-75, as pH differences are presented for P14.

ii.         Ambient air was normocapnic, did the authors observe hypocapnia in iNO treated pups?

iii.        Given the increased number and density of vessels in the cortex and white matter, the authors may speculate about oxygen transport/local oxygen concentrations and potential effects of related increase in local oxygenation, i.e. potential local hyperoxia triggering imbalances in redox proteins.

iv.        To improve data presentation, please add representative micrographs for MCT-1 and western blot images of assessed proteins were applicable, i.e. Figures 1-5 and 3 respectively.

v.         Did the authors also try to use other more common vascular markers like CD31 or van-Willebrand factor?

vi.        Though the discussion is balanced, the authors may speculate on long-term functional consequences of iNO treatment (without further injury), would the authors expect improved motor or cognitive function in this treatment paradigm.

vii.       The conclusion need to be rephrased. Unlike previous articles from the group, this study does not include an injury paradigm (hypoxia, hyperoxia or inflammation).  

minor comments:

 1.    As IJMS and MDPI endorse ARRIVE guidelines, the authors should address this in their methods. This group has published well-respected articles on effects of iNO treatment, did they perform a priori sample size calculations, what was the overall number of animals used in this study, how did they control for potential sex-differences, did they observe mortality, what kind of exclusion criteria/outlier identification have been performed (as numbers differ between groups, time points, and analysis) etc.?

2.     Appreciating the presentation of single data points in the figures, did they checked normal distribution were appropriate, i.e. t-test and one-way ANOVA?

3.    VEGF levels have been assessed in forebrain extracts, data for plasma concentrations are not provided, please adapt the methods accordingly.

4.    Fig. S3 would benefit from more detailed figure caption to make the figure more intuitive for a broader readership.

5.    Please harmonize writing of `nitrite/nitrate´ and use spaces in front of SI units.

Author Response

We thank the reviewers for her/his meaningful comments. Please, find our reply  in italics for each of these comments:

Major comments:

1. Did the authors observe differences in duration of pregnancies or litter size, was there a difference in breathing frequencies and cerebral blood flow in iNO-treated pups, as arterial CO2 was decreased in treated animals at earlier time points? Please check wording in lines 74-75, as pH differences are presented for P14.

R: We thank the reviewer for these valuable comments. Exposure to iNO was not associated with any difference in gestation length or litter size (birth weight is shown in Supplementary Figure S1). Unfortunately, respiratory rate was not routinely analysed in these animals. However, based on observations in humans, inhalation of nitric oxide is not associated with a change in breathing rate, particularly hyperventilation.

We have previously documented that exposure to inhaled nitric oxide has no effect on blood flow velocity in the basilar trunk as assessed by Doppler echography (Charriaut-Marlangue, C.; Bonnin, P.; Gharib, A.; Leger, P.-L.; Villapol, S.; Pocard, M.; Gressens, P.; Renolleau, S.; Baud, O. Inhaled Nitric Oxide Reduces Brain Damage by Collateral Recruitment in a Neonatal Stroke Model. Stroke 2012, 43, 3078-3084, doi:10.1161/STROKEAHA.112.664243)

We have rephrased the paragraph on gazometry according to Supplementary Table T1 (li 79-82), and added a precision regarding the lack of effect on blood flow velocities in the basilar trunk with reference to the previous article (li 82-85).

2. Ambient air was normocapnic, did the authors observe hypocapnia in iNO treated pups ?

R: We only observed a significant hypocapnia following no exposure at P1 but neither at P7 nor P14 (supplemental table S1).

3. Given the increased number and density of vessels in the cortex and white matter, the authors may speculate about oxygen transport/local oxygen concentrations and potential effects of related increase in local oxygenation, i.e. potential local hyperoxia triggering imbalances in redox proteins.

R: We thank the reviewer for this important comment. Because the hemoglobin level was slightly higher at P7 after NO exposure compared to the control, we cannot rule out that increased oxygen delivery may play some role in the increased vessel density in the cortex and white matter. This precision was added in the discussion (li 206-208).

4. To improve data presentation, please add representative micrographs for MCT-1 and western blot images of assessed proteins were applicable, i.e. Figures 1-5 and 3 respectively.

R: We have added micrographs of MCT-1 immunolabelling in the Figure 3. Regarding Figures 5 and 6, we should add eight and sixteen micrographs of MCT-1 immunolabelling, respectively. We are concerned that this may affect the readability of the figures and we defer to the editor's decision. We have added Western Blot Images in front of histograms 3-D and 3-C, as requested.

5. Did the authors also try to use other more common vascular markers like CD31 or van-Willebrand factor?

R: MCT1 results were confirmed using tomatolectin immunolabeling (TL 1:500, Vector Burlingame, CA), however due to the better specificity of MCT1 for mature vessels, subsequent vessel quantification was based on the latter.

We have added quantification and representative microphotographs at P14 in FIGURE 1-D and 1-E, and completed the methods and supplemental table S2 adequately.

6. Though the discussion is balanced, the authors may speculate on long-term functional consequences of iNO treatment (without further injury), would the authors expect improved motor or cognitive function in this treatment paradigm.

R: We thank the reviewer for this important comment. In the present experiments, we did not use a model of brain injury. We did not observe an increase in vascular development, but rather a shift in angiogenesis, as the vessel density usually observed at P21 in controls was already reached at P14 in animals exposed to iNO. We do not expect any improvement in motor or cognitive performance following this shift in vascular development. In a previous experiment, using the same animal model, we documented that exposure to 5 ppm NO exposure had no effect on three behavioral tasks: open field, new object, moving object, assessed at P60. (ref : Olivier, P.; Loron, G.; Fontaine, R.H.; Pansiot, J.; Dalous, J.; Thi, H.P.; Charriaut-Marlangue, C.; Thomas, J.-L.; Mercier, J.-C.; Gressens, P.; et al. Nitric Oxide Plays a Key Role in Myelination in the Developing Brain. J Neuropathol Exp Neurol 2010, 69, 828–837, doi:10.1097/NEN.0b013e3181ea5203.)

7. The conclusion need to be rephrased. Unlike previous articles from the group, this study does not include an injury paradigm (hypoxia, hyperoxia or inflammation).

R: We agree with the reviewer’s comment. We have rephrased the conclusion as follows:

“In conclusion, this study provides new insights into the biological basis of the effect of iNO in the developing brain. Our data support an effect of iNO on angiogenesis in the developing brain, through blood carriers, leading to the modulation of the cGMP pathway and, more specifically, regulated by TSP-1. This effect may explain, at least partly, the effect of iNO on myelination and its neuroprotective properties previously documented in rodent models of perinatal brain injury.” (li 254-259)

Minor comments:

1. As IJMS and MDPI endorse ARRIVE guidelines, the authors should address this in their methods. This group has published well-respected articles on effects of iNO treatment, did they perform a priori sample size calculations, what was the overall number of animals used in this study, how did they control for potential sex differences, did they observe mortality, what kind of exclusion criteria/outlier identification have been performed (as numbers differ between groups, time points, and analysis) etc ?

R: We thank the reviewer for these important comments. We have updated the methods according to the ARRIVE 2.0 guidelines.

About the a priori sample size : A priori sample size calculation indicated a minimum of 6-8 animals per group. Since exposure started on the last theoretical day of gestation, we assigned one to two pregnant rats and their litter, (usually at least twelve pups) per experimental condition, assessing one to two timepoints per litter. (li 273-276)

 About the total number of animals: Overall, two hundred and fifty rat pups were used in this study. (li 283-284).

On potential sex differences: We did not consider the sex effect in this article, as we had previously studied it without identifying any sex difference (Olivier, P.; Loron, G.; Fontaine, R.H.; Pansiot, J.; Dalous, J.; Thi, H.P.; Charriaut-Marlangue, C.; Thomas, J.-L.; Mercier, J.-C.; Gressens, P.; et al. Nitric Oxide Plays a Key Role in Myelination in the Developing Brain. J Neuropathol Exp Neurol 2010, 69, 828–837, doi:10.1097/NEN.0b013e3181ea5203)

 About mortality: We did not observe any mortality induced by exposure to nitric oxide. (li 284-285)

 About exclusion criteria and outliers identification: No a priori exclusion criteria were defined. Outliers were identified using the Grubb method. (li 355-356)

2. Appreciating the presentation of single data points in the figures, did they checked normal distribution were appropriate, i.e. t-test and one-way ANOVA?

R: When appropriate, normality of the distribution was checked using a D'Agostino-Pearson test or a Shapiro-wilk test, depending on the number of values (li 356-357).

3. VEGF levels have been assessed in forebrain extracts, data for plasma concentrations are not provided, please adapt the methods accordingly.

R: We have corrected it in the methods (li 325).

4. Fig. S3 would benefit from more detailed figure caption to make the figure more intuitive for a broader readership.

R: We have redesigned figure 3 for better clarity.

5. Please harmonize writing of `nitrite/nitrate´ and use spaces in front of SI units.

The writing of nitrite/nitrate was homogenized and one space was systematically added in front of SI units.

Reviewer 2 Report

This study provides new insight into the biological basis of the effect of inhaled NO in the developing brain using rats as a model system. The authors provide a convincing link between inhaled NO and angiogenesis via the modulation of the cGMP pathway by TSP-1. The presented data and methods substantiate the conclusions and interpretation. Overall it is well written and deserves publication, however, I like to address minor shortcomings as stated below.

Figure 1: Please add a note to C) Density P14 cortex identical to B)

Line 71, to improve understanding, please add/extend the information on the two quantitative assessments, in addition to the provided reference to the methods.

Title 2.4 It would be appreciated if you could reformulate the title to provide a statement (similar to 2.3)

In the Discussion, you state: „In the present study, while exposure to iNO induced a rapid and significant increase in circulating nitrate/nitrite, it did not result in any increase in brain S-nitrosothiols at P1 or P7.“ Preferably, please show the data on S-nitrosothiols, as it is missing within the manuscript. Alternatively, it should be omitted from the discussion.

Author Response

We thank the reviewer for those valuable comments. Pleas find our reply in italics after each of them.

Figure 1: Please add a note to C) Density P14 cortex identical to B)

R: We have homogenized legends of 1-B and 1-C          

Line 71, to improve understanding, please add/extend the information on the two quantitative assessments, in addition to the provided reference to the methods.

R: We have added information about these two methods of quantification in this paragraph (li 74-78)

Title 2.4 It would be appreciated if you could reformulate the title to provide a statement (similar to 2.3)

R: We rephrased the title 2.4 : “Inhaled NO effects in the developing brain are not mediated by NO-related angiogenic factors”

In the Discussion, you state: „In the present study, while exposure to iNO induced a rapid and significant increase in circulating nitrate/nitrite, it did not result in any increase in brain S-nitrosothiols at P1 or P7.“ Preferably, please show the data on S-nitrosothiols, as it is missing within the manuscript. Alternatively, it should be omitted from the discussion.

R: We have followed the reviewer’s recommendation and omitted this from the discussion for clarity.

Reviewer 3 Report

In the article "", Loron and colleagues test the hypothesis that inhaled nitric oxide (iNO) affects angiogenesis in grey and white matter of the cingulate cortex of neonate rats. Defining the effects and mechanisms of action of iNO in the lung and other organs have relevance for the treatment of human neonate pathology, which unfortunately remains poorly understood despite significant advances in animal research.

The authors report the results of experiments where inhaled nitric oxide was delivered to rat pups and dams during the first week of postnatal life and the effects were assessed using a battery of tests, including immunohistochemistry, PCR, Western blotting, and chemical analysis of blood. The main finding is that iNO promotes angiogenesis likely through the production of cGMP by soluble guanlyate cyclase, downstream of Thrombospondin1, CD42 and CD36. The results are impactful, since they add to previous findings from this group and others on the developmental effects of iNO in rodents.

Overall the study has a good experimental design and the results of experiments are clearly presented. Here are some suggestions to the authors:

Abstract, Line 18: replace "advancement in the developmental program" with a better descriptor like "shift". If endogenous NO does not trigger the canonical angiogenic response, then inhaled NO is acting artificially and not necessarily affecting a program of development. If a program is involved, then what program components are changed by exposure to iNO? Another reason to use "shift" instead of advancement is based on the results reported in Fig. 5B where one could argue that iNO delays the increase in TSP1 between P7 and P14. Information about TSP1 expression is missing at P21.

Abstract Line 21: I suggest to replace "NO remote effects in" with "transporting NO to the brain". The latter is more precise for the non expert reader.

Introduction, Line 38 add: "(BPD)" after bronchopulmonary dysplasia.

Line 47: "blow" should be "blood".

Results. I suggest the authors make a more consistent use of abbreviations. For example, "NO 20 ppm" is used inconsistently and additional abbreviations are introduced in figures 3-5 to refer to the same variable: "NO5", "NO 5ppm".

Results Line74 and 75. It is not clear what is the rationale for doing these measurements. Please clarify.

Line 85. Please clarify if E21 is the same as P0. Was i.p. injection at E21 in the pup, or the dam?

Figure 4 panel C. If Elisa VEGF was normalized, then what do units of pg/ml refer to?

Is supplemental figure S3 an author proposal?, or is there a reference from the published literature?

Lines 141, 156, 173. I would suggest deleting "remote" for clarity of presentation.

Conclusion Lines 243-244. Why not make a more clear statement that your data supports a model where iNO is transported to the brain through blood carriers, potentially explaining its effects on vascular and myelin development and brain repair?

Materials and methods, Lines 257-258. I ti snot clear what was done to the dams. Were the iNO exposed pups cross fostered to control dams, and the control pups to iNO exposed dams? Please clarify and explain rationale.

From the above comment, it is not clear if the exposure to iNO was continuous or interrupted. Please clarify.

It i not clear how many total animals were used in the study. Animal numbers are not given in the figures. What do symbols mean in the figures: individual animals? samples from different animals?

Author Response

We thank very much the reviewer for her/his valuable comments. Please, find our reply to each of them in italics:

Abstract, Line 18: replace "advancement in the developmental program" with a better descriptor like "shift". If endogenous NO does not trigger the canonical angiogenic response, then inhaled NO is acting artificially and not necessarily affecting a program of development. If a program is involved, then what program components are changed by exposure to iNO? Another reason to use "shift" instead of advancement is based on the results reported in Fig. 5B where one could argue that iNO delays the increase in TSP1 between P7 and P14. Information about TSP1 expression is missing at P21.

R: We thank the reviewer for this valuable comment. We agree with the use of "shift" instead of "advancement" and have made the change in the text (li 18).

TSP1 expression was not assessed at P21 because vessel density was the same in controls and NO-exposed animals.

Abstract Line 21: I suggest to replace "NO remote effects in" with "transporting NO to the brain". The latter is more precise for the non expert reader.

R: We agree with this important comment. We have followed this suggestion. And homogenized throughout the manuscript.

Introduction, Line 38 add: "(BPD)" after bronchopulmonary dysplasia.

R: We have added this abbreviation.

Line 47: "blow" should be "blood".

R: We have corrected it.

Results. I suggest the authors make a more consistent use of abbreviations. For example, "NO 20 ppm" is used inconsistently and additional abbreviations are introduced in figures 3-5 to refer to the same variable: "NO5", "NO 5ppm".

R: We have checked the consistency of abbreviations and used “NO 20 ppm” or “NO 5 ppm”.

Results Line74 and 75. It is not clear what is the rationale for doing these measurements. Please clarify.

R: We hope we have clarified this paragraph using the following sentence : “ As exposure to a prolonged or high level of iNO may induce methemoglobinemia and impair oxygen delivery to tissues, we have analyzed arterial blood gas in rat pups. No statistical difference was detected in methemoglobin and PaO2 between animals exposed to room air or iNO (Supplemental Table S1)

Line 85. Please clarify if E21 is the same as P0. Was i.p. injection at E21 in the pup, or the dam?

R: Thank you for your comment: the manuscript lacks consistency on this issue. We used “E21” as the last day before the expected delivery (which usually occurs overnight). All experiments (iNO exposure or i.p. injection) lasted from E21, to P7. At E21, i.p. injection was performed on the dam, then on the pups from P0. We have corrected this confusion between E21 and P0 whenever necessary in the manuscript (li 268-269).

Figure 4 panel C. If Elisa VEGF was normalized, then what do units of pg/ml refer to?

R: Thank you for your accurate observation: The VEGF concentration was not "normalized" but "adjusted" to the protein concentration (measured in mg/ml) in the sample. This has been corrected in the legend.

Is supplemental figure S3 an author proposal? or is there a reference from the published literature?

R: Supplemental figure S3 is proposed by the authors and the revised version is now fully original.

Lines 141, 156, 173. I would suggest deleting "remote" for clarity of presentation.

R: We have removed these occurrences of the word “remote” accordingly.

Conclusion Lines 243-244. Why not make a more clear statement that your data supports a model where iNO is transported to the brain through blood carriers, potentially explaining its effects on vascular and myelin development and brain repair?

R: We thank the reviewer for this proposition. We have rephrased the conclusion as follows:

“In conclusion, this study provides new insights into the biological basis of the effect of iNO in the developing brain. Our data support an effect of iNO on angiogenesis in the developing brain, through blood carriers, leading to the modulation of the cGMP pathway and, more specifically, regulated by TSP-1. This effect may explain, at least partly, the effect of iNO on myelination and its neuroprotective properties previously documented in rodent models of perinatal brain injury.”(li 254-259)

Materials and methods, Lines 257-258. I ti snot clear what was done to the dams. Were the iNO exposed pups cross fostered to control dams, and the control pups to iNO exposed dams? Please clarify and explain rationale.

From the above comment, it is not clear if the exposure to iNO was continuous or interrupted. Please clarify.

R: Control and exposed rat pups from the same litter were continuously exposed to room air or nitric oxide, respectively. Because birth naturally occurs at night, exposure was initiated on the last day of gestation (E21) so that animals were exposed to their experimental condition immediately after birth. To control for any maternal-mediated effects (e.g., milk), we rotated dams between cages twice a week without moving the rat pups.

We have rephrased the Methods accordingly (li 271-276).

It i not clear how many total animals were used in the study.

R: We thank the reviewer for this important question. Two hundred and fifty rat pups were used in this study. This information is now mentioned in the methods (li 283).

Animal numbers are not given in the figures. What do symbols mean in the figures: individual animals? samples from different animals?

R: Thank you for this important question. The number of animals is not shown in the figures because each "dot" symbol superimposed on the columns corresponds to a single animal.

Round 2

Reviewer 1 Report

Many thanks for the revised version, which improved the manuscript substantially. 

Three very minor points can be addressed in the production process of this manuscript.

1. please double-check main text for typos, Figure caption 1A refers "to two pictures left" but might refer to "insets", in line 143 there is an misplaced "n", and please harmonize notation of "CD36".

2. regarding representative images for MCT-1 and western blot, I also would leave this to the editor if pictures requested can be depicted in an additional supplementary file not disturbing readability.

3. Appreciating new suppl. figure S3, a clear description related to the findings of this study in the figure legend might improve readability to a broader readershipt

Author Response

We thank the reviewer for these comments.
We have double-checked the manuscript for typos, with the help of a native English speaker.

The legend of the figure 1-A has been refined.

The CD36 notation has been harmonized (li 25, 136, 140, 177, 180, 293).

The legend of the supplemental figure S3 was rephrased as: “ Schematic representation of soluble guanylate cyclase/cGMP pathway leading to intracellular effects of exogenous inhaled NO. iNO is remotely delivered to the endothelial cells of the developing brain using blood carriers. Diffusing into the cytoplasm, iNO promotes angiogenesis binding to the soluble Guanylate Cyclase (sGC), modulating cGMP production and down-regulating TSP-1 expression.”